# Applications of Genome-Editing Technologies for Type 1 Diabetes

**DOI:** 10.3390/ijms25010344

**Published:** 2023-12-26

**Authors:** Rana El Nahas, Mohammad Ameen Al-Aghbar, Laura Herrero, Nicholas van Panhuys, Meritxell Espino-Guarch

**Affiliations:** 1Laboratory of Immunoregulation, Translational Medicine, Sidra Medicine, Doha P.O. Box 26999, Qatar; ranaelnahhas@gmail.com (R.E.N.); malaghbar@sidra.org (M.A.A.-A.);; 2Department of Biochemistry and Physiology, School of Pharmacy and Food Sciences, Institute of Biomedicine of the University of Barcelona (IBUB), 08028 Barcelona, Spain; lherrero@ub.edu

**Keywords:** Type 1 diabetes, genome-editing, CRISPR–Cas9, autoimmune disease, β-cell, therapy

## Abstract

Type 1 diabetes (T1D) is a chronic autoimmune disease characterized by the destruction of insulin-producing pancreatic β-cells by the immune system. Although conventional therapeutic modalities, such as insulin injection, remain a mainstay, recent years have witnessed the emergence of novel treatment approaches encompassing immunomodulatory therapies, such as stem cell and β-cell transplantation, along with revolutionary gene-editing techniques. Notably, recent research endeavors have enabled the reshaping of the T-cell repertoire, leading to the prevention of T1D development. Furthermore, CRISPR–Cas9 technology has demonstrated remarkable potential in targeting endogenous gene activation, ushering in a promising avenue for the precise guidance of mesenchymal stem cells (MSCs) toward differentiation into insulin-producing cells. This innovative approach holds substantial promise for the treatment of T1D. In this review, we focus on studies that have developed T1D models and treatments using gene-editing systems.

## 1. Introduction

Type 1 diabetes (T1D) has a global prevalence of approximately 9.5%, with an incidence rate of 15 cases per 100,000 people [1]. Despite first being recognized as early as 1500 BC [2], T1D was universally fatal until the 1920s, when insulin was first isolated and administered to patients [3]. After decades of research into the cellular and molecular pathology of T1D, it is now defined as an autoimmune disease characterized by progressive pancreatic β-cell loss resulting in insulin deficiency and hyperglycemia [4]. Over the years, numerous treatment methods have been introduced for T1D. In addition to insulin replacement therapy, β-cell replacement using either allogenic solid organ pancreas or islet transplantation has demonstrated the potential to reverse T1D but necessitates lifelong immunosuppression to prevent graft rejection [5]. There has been notable success in preventing the development of T1D in mouse models using clustered regularly interspaced palindromic repeats (CRISPR) paired with CRISPR-associated protein 9 (CRISPR–Cas9)-mediated genetic modifications, for example, by converting B lymphocytes to a disease-inhibitory CD73+ regulatory state [6] or precisely guiding mesenchymal stem cells (MSCs) toward differentiation into insulin-producing cells (IPCs) [7]. In this review, we aim to provide an overview of the currently reported gene-editing models and interventions tested to prevent the onset of and treat T1D.

### 1.1. Diabetes Etiology

Understanding the etiology of diseases is crucial to gaining insight into their origins and pathologies and forming a fundamental basis for the development of effective therapeutic interventions. T1D is an autoimmune disease that results from the destruction of pancreatic β-cells, necessitating lifelong dependence on insulin [8]. While significant progress has been made toward understanding the cellular pathology and refining strategies for disease management using conventional techniques, the advent of next-generation sequencing and the application of genome-wide association studies (GWAS) have revolutionized the understanding of the disease. By as early as 2010, genetic mapping and gene–phenotype studies had uncovered 10 key genes associated with T1D, including those responsible for encoding major histocompatibility complex (MHC) human leukocyte antigen (HLA) class II and class I antigen-presenting molecules, preproinsulin (INS) in the thymus, cytotoxic T-lymphocyte antigen-4 (CTLA-4) in Treg cells, and cytokines such as IL-2, IL-7, and IL-10, which are all believed to be causative factors in the development of T1D [9]. T1D is a polygenic autoimmune disease, and more than 60 genetic loci with common variants that contribute to disease susceptibility have been identified in humans, with most having a low individual risk score [10]. In both mice and humans, the MHC region plays a substantial role, accounting for a significant portion of the risk, with approximately 40–50% attributed to specific class II HLA alleles, along with the combined effects of other loci [10]. The cellular and molecular pathology of T1D indicates that it is primarily an autoimmune disease driven by T cells [11]. It was shown that in T1D, insulin and its precursor, preproinsulin, are major self-antigens targeted by T cells [12], and it was reported that patients with diabetes carrying the PTPN22^R620W^ genetic variant showed higher levels of insulin autoantibodies (IAAs) [13]. Studies have also found a connection between a rapid degradation rate in the immunoproteasome and inflammation, leading to autoimmunity and the destruction of pancreatic β-cells mediated by autoreactive T cells [11]. However, the cause of T1D remains uncertain in many individuals, with some studies suggesting a potential link to low-grade enteroviral infection within pancreatic islets, which is supported by the presence of enterovirus in the pancreatic islets of T1D patients [14,15]. Others have suggested a bacterial role in the development of T1D, in which butyrate-producing lactate-utilizing bacteria disrupt mucin synthesis in the gut, leading to autoimmune disorders, particularly T1D [16]. Genetic predisposition involving the 2′-5′-linked oligoadenylate (2′-5′A) pathway has also been identified, potentially contributing to dysregulation of the innate antiviral immune system in T1D [17]. This suggests the possible involvement of the 2′-5′A pathway and other components of the innate antiviral immune system in β-cell autoimmunity [17].

### 1.2. Gene-Editing Technologies and CRISPR–Cas Systems

Gene-editing-based therapies involve altering the gene sequence in a targeted manner to address diseases. This method works by modifying specific genes by replacing a faulty gene with a healthy version, deactivating a malfunctioning disease-related gene, or introducing a new or modified gene to assist in treating the illness. Various techniques, such as plasmid DNA, viral vectors, CRISPR–Cas, transcription activator-like effector nucleases (TALENs), and zinc-finger nucleases, are employed in gene editing to precisely modify the genetic material [18].

CRISPR–Cas was initially identified as a prokaryotic defense system present in bacteria and archaea and has since become a molecular tool for precise gene editing. At its core, CRISPR–Cas employs guide RNAs to direct the Cas nuclease to specific genomic sequences, inducing targeted DNA double-strand breaks. These breaks trigger the cell’s endogenous repair pathways, notably non-homologous end joining (NHEJ) and homology-directed repair (HDR). NHEJ operates by directly ligating broken DNA ends, often leading to error-prone repair, resulting in insertions or deletions. In contrast, HDR, which is facilitated by CRISPR–Cas through the provision of repair templates, such as donor DNA sequences, enables precise genomic alterations [19].

The importance of modifications in CRISPR gene editing is emphasized, particularly in overcoming limitations associated with the commonly used SpCas9 enzyme. One key challenge is posed by the specific PAM (protospacer adjacent motif) requirement of SpCas9, which restricts its applications for precise genome editing. To address this, various Cas9 orthologs, such as high-fidelity variants and those with different PAM sequence preferences, have been explored. These modifications, including the use of Cas12 orthologs with broader PAM recognition sites, signify ongoing efforts to enhance the specificity and versatility of CRISPR technologies for more effective and precise genome editing. Exploring the functionality of different CRISPR–Cas nucleases highlights their diverse roles in genome editing. Table 1 summarizes the distinctive roles and features of Cas13, Cas9, Cas12, Cascade–Cas3, and Cas7–11 nucleases [19].

## 2. T1D In Vivo Models

In vivo models for studying T1D are crucial for understanding the pathogenesis of the disease and testing potential treatments. Animal models, such as mice, rats, dogs, zebrafish, and pigs, have been used to replicate various aspects of human T1D. 

The BioBreeding rat model possesses a frameshift mutation in the immune-associated nucleotide-binding protein gene, *Ian4*, and spontaneously develops diabetes, marked by lymphopenia and diabetic retinopathy [20]. 

Canine models, including specific breeds like the Samoyed and Keeshond, can spontaneously develop diabetes similar to humans [21]. 

Zebrafish offer a versatile model for diabetes research, via diverse protocols, such as drug-induced β-cell destruction and transgenic lines expressing toxin genes. The *Vhl* zebrafish model carries a mutation in the von Hippel–Lindau tumor suppressor gene, resulting in increased blood vessel formation and upregulation of hypoxia-inducible factor. This leads to T1D as well as vascular leakage, macular edema, retinal detachment, and severe neovascularization [22]. The Tg(ins:egfp-nfsB) zebrafish model expressing the EGFP–NTR fusion protein in β-cells has an effective system for β-cell ablation and regeneration [22].

Pigs are also a valuable model for T1D research due to their physiological and anatomical similarities to humans, allowing researchers to study various aspects of diabetes, including pancreatic islet transplantation, insulin production, glucose metabolism, and immune responses. These animal models enable investigations into transplantation techniques and immunological reactions associated with islet transplantation, which is a potential therapeutic approach for T1D. CRISPR–Cas9 technology has allowed the generation of pigs with modified immune-modulatory genes to make pig islets less prone to degradation mechanisms and enhance their compatibility with the recipient’s immune system. This advance shows promise in overcoming hurdles, such as blood-mediated inflammatory reactions and immune incompatibility in xenotransplants [23].

Mouse models, including pathogen-induced models, spontaneous models, transfer models, and pharmacological (streptozotocin-induced diabetes) models [24], are the most widely used to study T1D. The non-obese diabetic (NOD) mouse model is a spontaneous model of T1D that shares highly analogous characteristics with the onset of T1D in humans and is among the most widely used models. NOD mice, originally derived from cataract-prone outbred Jcl:ICR mice, have been shown to be essential for studying T1D, [25] as they carry the H-2g7 haplotype. In particular, the MHC Class II molecule, I-Ag7, which significantly contributes to the development of T1D due to its unique peptide binding characteristics, potentially leads to a diverse range of autoreactive T cells. Additionally, over 20 non-MHC loci have been associated with diabetes pathogenesis in NOD mice, with genes including IL2 and CTLA-4 playing essential roles, highlighting the complex and multi-step genetic pathways involved in the development of autoimmune diabetes in these mice [25]. NOD mice were recently used to study T1D, including the MHC haplotype and various insulin-dependent diabetes susceptibility (Idd) loci [10], which have deepened our understanding of the disease by providing detailed genetic and mechanistic insights into the development of T1D, offering a foundation for more targeted and effective approaches to diagnosis, treatment, and prevention of the disease. To facilitate the translation of knowledge gained from the study of these genetic and pathological pathways into clinical therapies, efforts have been made to “humanize” NOD mice by introducing genes relevant to T1D patients, creating a versatile platform for testing therapies tailored to a diverse population at risk of T1D [26].

### CRISPR–Cas9-Engineered T1D Mouse Models

CRISPR–Cas9 gene knockouts have revolutionized research by streamlining the generation of knockout animals, drastically reducing the time required to create genetically modified models [27]. This technology has facilitated the development of animal models for various diseases, enabling the study of genetic variations and accelerating the development and testing of treatments. To date, 13 genetically modified mice have been generated using CRISPR–Cas9 technology to create alterations in specific genes related to immune regulation or β-cell function to mimic certain aspects of T1D (summarized in Table 2).

***Ptpn22***: In 2016, a mouse model carrying a Ptpn22(R619W) mutation was the first T1D mouse model engineered using CRISPR–Cas9. A single amino acid mutation, R619W, was introduced into the endogenous protein tyrosine phosphatase nonreceptor type 22 (PTPN22) gene. This resulted in the creation of Ptpn22^R619W^ mutant mice, which led to an enhanced rate of spontaneous development of T1D, coincident with increased levels of IAAs in homozygous and heterozygous female Ptpn22^R619W^ mice [13].

**AID/RAD51 Axis**: Researchers developed an activation-induced cytidine deaminase (AICDA) gene knockout NOD mouse model (NOD.Aicda^−/−^) using CRISPR–Cas9 technology. B lymphocytes from wild-type NOD mice showed high levels of constitutive AID expression, while ablation of AID led to significant delays in T1D development but did not change the overall diabetogenic activity. However, combining inducible AICDA ablation with treatment with RAD51 inhibitor expanded the CD73+ B lymphocytes that exert regulatory activity, which effectively inhibited diabetogenic T-cell responses. The AID/RAD51 axis has subsequently been recognized as a potential target for a pharmacological approach with potential clinical translation. This approach aims to impede the onset of T1D by transforming B lymphocytes into a regulatory state characterized by the disease-inhibitory CD73+ phenotype [6].

**PTPN**: CRISPR–Cas9 technology was employed to selectively inactivate two specific genes encoding protein tyrosine phosphatases (PTPN6 and PTPN1) in β-cells and islets. This genetic modification allowed researchers to investigate the roles of these proteins in the regulation of cytokine signaling and β-cell death in the context of T1D. It was determined that PTPN6 negatively regulated TNF-α-induced β-cell death, while PTPN1 positively regulated IFN-γ-induced gene expression, contributing to autoimmune β-cell destruction. Additionally, inactivating PTPN1 through pharmacological methods protected β-cells from cytokine-induced cell death. Together, these findings highlight the non-redundant roles of PTPs in regulating cytokine signaling in β-cells during autoimmune diabetes [37].

**HLA class II variants**: CRISPR–Cas9 technology was employed to generate new mouse models by introducing human disease-associated HLA-A02:01 or HLA-B39:06 class I molecules into NOD.b2m2/2 mice, which are normally resistant to T1D. The development of T1D in these models depends on pathogenic CD8+ T-cell responses mediated by human class I variants. The use of these models allowed for the identification of novel β-cell autoantigens and the testing of targeted therapies. Further, NOD.b2m2/2 mice lack the nonclassical MHC I family members, including FcRn, necessary for antigen presentation and maintaining serum proteins, including IgG and albumin. MHC class I variants were removed using CRISPR–Cas9, creating NOD mice lacking classical murine MHC expression (cMHCI/II^−/−^) while retaining nonclassical MHC I molecule expression and FcRn activity. Transgenic expression of HLA-A2 or HLA-B39 restored pathogenic CD8+ T-cell development and T1D susceptibility, making them promising platforms for T1D therapy development [26].

***Nfkbid***: The development of T1D in both NOD mice and humans involves the expansion of autoreactive CD8+ T cells that recognize pancreatic β-cell peptides presented by common MHCI variants. Previous studies in NOD mice revealed that the common H2-Kd and H2-Db class I molecules (molecules that play a crucial role in presenting antigens to cytotoxic CD8+ T cells) lose their ability to induce thymic deletion of pathogenic CD8+ T-cell responses due to interactions with T1D susceptibility genes outside the MHC loci. Proximal chromosome 7 was identified as a contributor to this phenomenon, with elevated expression of *Nfkbid* gene variants. Using CRISPR–Cas9 technology, *Nfkbid* expression was attenuated in NOD mice, leading to improved negative selection of autoreactive CD8+ T cells. Surprisingly, ablation of *Nfkbid* expression accelerated T1D onset, which was associated with decreased regulatory T and B lymphocytes in NOD mice. This suggests that while enhancing thymic deletion of pathogenic CD8+ T cells, Nfkbid also plays a role in regulating T1D onset, potentially affecting the thymic development of additional immune cell populations, including altering the Treg repertoires [26].

**CNS1**: Deletion of conserved non-coding sequence 1 (CNS1) in the *Foxp3* locus leads to selective impairment of peripheral Treg (pTreg) generation without disrupting thymic Treg (tTreg) generation in a C57BL/6J background. CRISPR–Cas9-mediated deletion of the CNS1 locus of FOXP3 in NOD mice was used to generate NOD CNS1^−/−^ mice. There was a small but significant decrease in RORγt+ Tregs and a corresponding increase in Helios+ Tregs. However, the deletion of CNS1 did not affect the development of T1D or glucose tolerance. Further, the proportions of autoreactive Tregs and conventional T cells within the pancreatic islets were unchanged. Together, these results indicate that pTregs, which are dependent on the Foxp3 CNS1 region, are not essential to the development of T1D in NOD mice, suggesting that tTregs are critical in preventing and reversing autoimmunity [30].

***Rnls***: The application of genome-scale CRISPR–Cas9 screening by Cai et al. identified *Rnls* (a gene associated with human T1D in an earlier GWAS [31]) as a modifier of β-cell survival in the NOD mouse model. Mutation in the *Rnls* gene protected both the NIT-1 cell line and primary NOD β-cells against autoimmune destruction by diminishing immune recognition by autoreactive T cells. Additionally, the effects of Rnls deletion were reproduced by oral administration of a previously U.S. Food and Drug Administration (FDA)-approved drug, pargyline, presenting a potential pharmacological agent that could be repurposed as a preventive therapeutic for T1D [31].

**STING**: CRISPR–Cas9 was used to target the stimulator of IFN genes (STING) for deletion in NOD mice, generating STING^−/−^ NOD mice. Here, the STING deficiency partially reduced the level of type I interferon gene signature present in islets during the onset of T1D to a level that was insufficient to inhibit insulitis. Interestingly, STING-deficient NOD mice have increased numbers of islet-specific glucose 6-phosphatase catalytic subunit-related protein IGRP206-214-specific CD8+ T cells and a higher incidence of spontaneous diabetes compared with wild-type NOD mice. Moreover, splenocytes from STING-deficient mice induced diabetes more rapidly when transferred to irradiated NOD recipients. This indicates that while STING may contribute to the type I interferon gene signature, its knockout revealed an unexpected role in suppressing diabetogenic T cells in NOD mice [32].

***Dusp10***: Following the breeding of two sublines of NOD mice with high (NOD/Nck^H^) and low (NOD/Nck^L^) T1D incidence, whole-genome sequencing identified relatively few subline-specific genetic variants. Automated meiotic mapping revealed an increased susceptibility to T1D in the NOD/Nck^H^ mice due to a recessive missense mutation in the dual specificity phosphatase, *Dusp10,* gene. This mutation was confirmed to increase disease incidence when introduced using CRISPR–Cas9 into NOD/Nck^L^ mice, as Dusp10 mutation led to downregulation of the type I interferon signature present in islet cells. This potentially offers protection against autoimmune attacks and highlights how de novo mutations, similar to rare human susceptibility variants, can significantly alter T1D phenotypes [10].

**CXCL10**: Enhancing the viability of stem cell-derived islets (SC-islets) for potential use in treating insulin-dependent diabetes without the need for encapsulation or immunosuppressive drugs is an attractive treatment strategy. Single-cell RNA sequencing and whole-genome CRISPR screening were conducted to assess immune system interactions in SC-islets, particularly in regard to allogenic peripheral blood mononuclear cells. Activation of JAK/STAT type II IFN pathways in SC-islets was determined to be a leading modulator of early and late inflammatory response events both in vitro and in vivo. Through CRISPR-based interventions, modulators of JAK/STAT pathway members were manipulated, specifically depleting chemokine ligand 10 (CXCL10). This resulted in improved SC-islet survival when exposed to immune attacks and provided valuable insight into immune-related challenges in SC-islet transplantation, unraveling potential gene editing targets to enhance their success in therapeutic applications [33].

**Lentiviral vector-induced tolerance**: The T-cell repertoire, which makes NOD mice prone to T1D, can be altered through intrathymic administration of lentiviral vectors (LVs) designed for gene transfer to thymic epithelial cells (TECs). The presentation of LV-introduced antigens by TECs, including the immunodominant portion of ovalbumin and InsulinB9-23R22E, during T-cell development was able to establish central tolerance to insulin and effectively prevent autoimmunity [34].

**FOXP3 locus**: Introduction of the strong promoter element, MND (myeloproliferative sarcoma virus enhancer, negative control region deleted, dl587rev primer-binding site substituted) into the endogenous FOXP3 locus by CRISPR homology-directed repair, CRISPR HDR-mediated gene editing, facilitated stable FOXP3 expression in human CD4+ T cells. FOXP3 expression resulted in the robust production of engineered cells with a Treg phenotype and suppressive function (eTregs). Human islet-specific eTregs were generated using LVs encoding islet-specific T-cell receptors. These antigen-specific cell products were able to suppress both direct and bystander effector T-cell (Teff) responses through a variety of mechanisms under both in vitro and in vivo conditions, where they inhibited the activation and production of inflammatory molecules by islet antigen-recognizing Teffs, as well as bystander Teffs that target alternate islet antigens. Adoptive transfer of islet-specific eTregs into murine hosts resulted in their homing to the pancreas and prevention of T1D through inhibition of Teff inflammatory capacity [35].

**TCRa**: A combination of CRISPR–Cas9-mediated knockout of the endogenous TCRa chain gene, TRAC, and LV-mediated TCR gene transfer into primary human CD8+ T cells revealed antigen-specific CD8+ T-cell receptor specificity and function in T1D. Knockout enhanced de novo TCR pairing, leading to increased peptide–MHC Dextramer staining and elevated activation and effector function markers, such as granzyme B and INFγ. The engineered CD8+ T cells demonstrate enhanced cytotoxicity towards an HLA-A*0201+ human β-cell line, suggesting the potential for altering T-cell specificity for mechanistic analysis and downstream therapeutic applications in T1D [36]. 

While essential for preclinical research, these models have limitations and are different from human disease, emphasizing the need for comprehensive studies that integrate findings from animal models with human clinical observations to advance our understanding and treatment of T1D.

## 3. β-Cell Replacement

### 3.1. Insulin Production

Early trials utilizing adenoviruses and retroviruses as viral vectors aimed to achieve sustained insulin gene expression in the liver, offering promise for T1D treatment [38]. However, challenges, such as delayed kinetics of transcriptional regulation and risk of hypoglycemia, persist. Synthetic promoters incorporating glucose-responsive elements and pharmacological control systems have been investigated to enhance regulation [39]. Insulin gene therapy, which targets non-exocrine tissues for insulin expression, can be complemented by an alternative approach—gene therapy-induced β-cell generation. The use of the key protein, PDX-1, demonstrated successful reprogramming of hepatocytes into insulin-secreting β-cell-like clusters. Co-expression with other transcription factors enhanced efficiency, suggesting potential for autologous β-cell replacement therapy. Another strategy involves inducing β-cell neogenesis through the activation of key regulators, offering promising alternatives to traditional insulin gene therapy [40]. An alternative surgical technique to reverse T1D through gene therapy involves isolating the liver and delivering a lentiviral vector carrying furin-cleavable human insulin (INS-FUR) to diabetic NOD mice. This method resulted in long-term transduction of hepatocytes, restoring normoglycemia for 150 days. The mice exhibited normal glucose tolerance, expressed β-cell transcription factors, and stored insulin in hepatic granules. Using this surgical approach, the gene therapy protocol achieved permanent reversal of T1D with normal glucose tolerance in NOD mice, presenting a promising therapeutic strategy [41].

### 3.2. Transplants

Pancreatic islet transplantation is an established therapy for T1D, and recent advances in islet isolation and transplant procedures have improved patient outcomes [42]. Pancreas and cadaveric islet transplantation, which are typically reserved for patients who are unresponsive to standard care or require kidney transplantation, face donor limitations and also require the use of immunosuppressive drugs [43]. Allogenic islet transplantation offers a safe alternative for patients with severe hypoglycemia or unstable blood sugar levels [44]. Additionally, intrahepatic islet transplantation effectively stabilizes diabetes and prevents complications [44]. Further, the use of transplanted purified human pancreatic islets in medically unstable T1D patients resulted in glycemic control, restoration of hypoglycemia awareness, and protection from severe hypoglycemia [45]. However, organ shortage, aggressive immunosuppression, and potential graft rejection remain obstacles in translating these advances to broad clinical use, emphasizing the need for future improvements focused on increasing efficacy and safety profiles [46]. Recently, the FDA approved Lantidra, the first allogeneic pancreatic islet cellular therapy made from deceased donor cells, for the treatment of T1D in adults who experience severe hypoglycemia despite intensive diabetes management. Lantidra is administered through a single infusion into the hepatic portal vein and has shown effectiveness in reducing or eliminating the need for insulin in some patients, but adverse reactions in others. This treatment should be considered after assessing the benefits and risks for each individual patient [47].

The shortage of donors for β-cell replacement is a major issue that could be resolved by xenotransplantation. The first transplantation from a porcine pancreas was performed in Sweden 30 years ago [48]. Currently, only encapsulated neonatal porcine islet xenotransplantation into the peritoneal cavity is performed clinically, since naked islet transplantation triggers an instant blood-mediated inflammatory reaction (IBMIR) that requires immunosuppression [49]. The use of genetically engineered pig donors in islet transplantation has improved issues related to IBMIR and autoimmunity [50], such as knocking out carbohydrate alpha 1,3 galactose (Gal) to avoid acute rejection [51]. The transgenic expression of human CD46 improved a xenotransplant outcome by limiting antibody-mediated rejection [52].

### 3.3. Stem Cells and Induced Pluripotent Stem Cell Therapy

Despite encouraging preclinical findings for islet xenotransplants, the clinical application of neonatal porcine islets faces challenges due to delayed insulin secretion resulting from immaturity and immunogenicity [53]. Multipotent MSCs are recognized for their pro-angiogenic, anti-inflammatory, and immunomodulatory attributes. Studies have underscored the considerable potential of co-culturing and co-transplanting islet cells with MSCs. Investigations have demonstrated amplified islet proliferation and maturation, increased insulin secretion, and improved graft survival, leading to enhanced graft outcomes [54]. However, co-transplantation of SC-islets offers advantages, as they provide a limitless cell source for therapy. 

The seminal 2006 study by Takahashi and Yamanaka demonstrated the ability to reprogram differentiated cells into a state similar to embryonic cells by introducing Yamanaka’s factors (Oct3/4, Sox2, Klf4, and c-Myc) under conditions used for embryonic stem cells (ESCs) [55]. Induced pluripotent stem cells (iPSCs) offer an exceptional platform for modeling and treating human diseases [56]. Breakthroughs in iPSC technology suggest that patient-specific iPSCs could serve as a replenishable source of autologous cells for cell therapy. A seven-stage protocol efficiently converts human ESCs into insulin-producing cells, offering a potential therapy for diabetes. Autologous diabetes cell therapy using patient-specific iPSC-islets, which contain the missing insulin-secreting β-cells, has the potential to eliminate the need for post-transplant immunosuppression [57]. Advances in protocols will facilitate the production of large batches of cells and cryopreservation of functional SC-β-cells [58]. However, there is a concern about de novo mutations in mitochondrial DNA, which can give rise to neoantigens, depending on the individual’s MHC genotype [59]. However, even when autologous iPSCs are found to be non-immunogenic, it remains a challenge to use them for β-cell replacement in T1D patients due to the pre-existing autoimmunity against islet antigens. This autoimmunity can destroy transplanted iPSC-derived β-cells, as evident in cases in which T1D is transferred between siblings post-bone marrow transplant and in the development of T1D following islet auto-transplantation [60,61,62]. Therefore, exposure to autologous islet antigens may trigger an immune attack against transplanted iPSC-derived β-cells.

## 4. Strategies for Protecting Transplanted β-Cells

Brusko et al. described a strategy for protecting transplanted β-cells from immune-mediated rejection involving a two-phase immunomodulatory approach that includes implantation of pancreatic progenitor cells and in vitro generation of stem cell-derived β-like cells, resulting in the creation of stem cell-derived β-like cells (sBCs). However, each phase has its own set of challenges. This approach is crucial for ensuring the long-term success of β-cell or islet transplantation in patients with established T1D. Current islet transplantation methods employ anti-rejection drugs and preconditioning, while next-generation therapies will incorporate novel experimental methods and adoptive cell therapies, utilizing Tregs to counter recurrent autoimmunity and establish enduring graft tolerance. Following host conditioning with conventional immunosuppressants, gene editing can be applied to shield sBCs from immune-mediated damage, and biomaterial-based techniques can provide physical protection and localized immunosuppression, ultimately enhancing graft survival [63]. 

## 5. Immunosuppressant Drugs

Immunosuppressant drugs that target T–B lymphocyte interactions, such as T-cell elimination (anti-CD3, teplizumab), B-cell elimination (anti-CD20, rituximab), and exploration of T-cell activation disruption (CTLA4/Fc-fusion, abatacept) have been investigated as strategies for protecting implanted β-cells from immune rejection (Figure 1). However, there are risks associated with global immune cell subset disruption, particularly in children [64]. Teplizumab is seen as a significant advance in T1D therapy. In five trials, it effectively delayed the decline in C-peptide levels, improved glycemic control in new-onset T1D, and reduced the incidence in at-risk relatives. Adverse effects were generally mild to moderate. While teplizumab represents progress in T1D treatment, further research is needed to explore screening, cost-effectiveness, treatment accessibility, optimal treatment duration, and potential combination therapies [65].

Although the immunogenicity of transplanted cells can currently be managed using traditional immunosuppressive medications, standard immunosuppressive regimens in clinical islet transplantation often also involve induction therapy using anti-thymocyte globulin or IL-2 receptor monoclonal antibodies, followed by maintenance treatment with immunosuppressive drugs, such as tacrolimus and sirolimus [43]. 

## 6. Engineered Immune Cells

Tregs are immunosuppressive T cells that are critical for the maintenance of tolerance in vivo. GWAS and transcriptional and functional analyses have highlighted Foxp3 as a key immunological defect in Tregs in T1D. Treg-based therapies are seen as potential solutions to rebalance the system and reverse autoimmunity. However, challenges exist due to the complexity of the function of Tregs and limited information about tissue-infiltrating Tregs in T1D [66].

Robust techniques have been developed to isolate and expand Tregs from T1D patients. In a phase 1 trial assessing safety, adults with T1D received ex vivo-expanded autologous CD4+CD127^lo/−^CD25^+^ polyclonal Tregs in four dosing cohorts. The expanded Tregs, which demonstrated enhanced functional activity and T-cell receptor diversity, were long-lived in the circulation, with up to 25% remaining at 1-year post-transfer. An immune study revealed transient increases in Tregs, which retained the Treg phenotype long-term. The study showed no adverse events, and C-peptide levels persisted for over 2 years in some individuals, supporting the development of a phase 2 trial to test the efficacy of Treg therapy [67]. Combination approaches for Treg cell transfer involve using IL-2 receptor and STAT5 agonists concurrently. Additional strategies include reducing intrinsic effector T-cell responses through anti-T-cell therapies like anti-thymocyte globulin or tumor necrosis factor (TNF) inhibitors. However, challenges arise due to the overlapping physiology between Tregs and effector T cells. For instance, TNF inhibitors, though effective in some autoimmune conditions, may paradoxically worsen diseases or trigger new autoimmune conditions. Similar challenges exist with IL-2 therapy, mTOR inhibitors, and other potential treatments, necessitating careful consideration of timing and specificity when inhibiting proinflammatory pathways during Treg adoptive cell transfer for organ transplant rejection and autoimmune diseases [68]. 

Redirecting polyclonal Tregs to generate and expand antigen-specific Tregs includes using chimeric antigen receptors (CARs) or engineered T-cell receptors. Engineered tissue antigen-specific Treg cells showed increased efficiency in preclinical models of autoimmune diseases and transplantation. TCR-transduced Tregs accumulate in targeted tissues during autoimmunity, exerting antigen-specific and bystander suppression [68]. A promising strategy for transplant tolerance was explored using Tregs engineered with an anti-HLA-A2 chimeric antigen receptor (A2-CAR). A2-CAR Tregs, which were designed to target donor-derived HLA, selectively accumulated in HLA-A2-expressing islets without impairing their function. A2-CAR Tregs delayed graft-versus-host disease in the presence of HLA-A2, demonstrating their potential for precision Treg therapies in achieving transplant tolerance [69]. A crucial epitope (B:9–23) in the NOD mouse model of T1D was identified, and a monoclonal antibody (mAb287) targeting the I-Ag7-B:9–23(R3) complex was targeted [70]. This complex involves the interaction between the I-Ag7 molecule, an MHC class II molecule specific to the NOD strain, and the peptide sequence B:9–23, which spans residues 9 to 23 of the insulin B chain. The (R3) notation denotes a specific register, highlighting the unique binding configuration of the B:9–23 peptide to the I-Ag7 molecule [71]. T1D onset was significantly delayed or prevented in prediabetic mice through weekly administration of mAb287. To enhance efficiency, CAR T cells (287-CAR) targeting the same I-Ag7-B:9–23(R3) complex were infused into young NOD mice. Diabetes onset was significantly delayed compared with controls, although protection declined over time. Mechanistic studies revealed selective homing of 287-CAR T cells with pancreatic lymph nodes, with the Ag7-B:9–23(R3) complex playing a central role. While effective, T1D development was only delayed, not prevented, by a single infusion of 287-CAR CD8+ T cells, indicating the need for studies to identify strategies to improve cell longevity [70]. Treg therapy is a promising approach, but chronic activation by the CAR system resulted in the loss of suppressive function by Treg exhaustion, and these issues need to be addressed [72].

B cells enhanced with lipopolysaccharide for improved regulatory function were engineered (e-B cells) with chimeric MHC-I or MHC-II molecules linked to antigenic peptides. In vitro and in vivo experiments showed that e-B cells inhibited CD8+ T-cell cytotoxicity and induced regulatory markers in CD4+ T cells. Additionally, e-B cells protected mice from autoimmune diabetes induced by the transfer of antigen-specific CD8+ and CD4+ T cells. The study underscores the potential of MHC–peptide chimeric e-B cells in interacting with pathogenic T cells and suggests their use in selectively targeting and regulating specific pathogenic cells for therapeutic purposes [73].

Furthermore, it is established that genetic modification of β-cells using CRISPR–Cas9 technology could be used to protect implanted cells from autoimmunity through techniques including co-secretion of IL-10 or replacing HLA molecules, both of which show potential for preventing immune-dependent rejection [74]. 

## 7. Conclusions

Gene editing, specifically CRISPR–Cas9 technology, has emerged as a promising treatment approach for a range of autoimmune diseases, including but not limited to rheumatoid arthritis, inflammatory bowel disease, systemic lupus erythematosus, multiple sclerosis, T1D, psoriasis, and type 1 coeliac disease [35]. In November 2023, CRISPR-based treatment was approved in the UK in a global first. The UK’s Medicines and Healthcare products Regulatory Agency granted regulatory approval for the CRISPR gene editing treatment, known as Casgevy, marking the first approval of a medical treatment involving CRISPR worldwide. Casgevy was designed to treat sickle cell disease and β-thalassemia, which are genetic conditions caused by errors in hemoglobin genes, with no universally successful treatment currently available for either disorder [75].

In the context of T1D, the ultimate goal for therapy may involve a combination of approaches. One such approach could be the use of iPSC-derived β-cell transplants to replace the IPCs that are destroyed in T1D, together with a gene therapy approach to control levels of proinflammatory signaling molecules and prevent infiltration of lymphocytes into the transplanted β-cell islets. This dual approach aims to protect the newly functional β-cells from autoimmune attacks, ultimately improving the long-term management of T1D and offering the possibility of a cure or significant disease control.

## Figures and Tables

**Figure 1 ijms-25-00344-f001:**
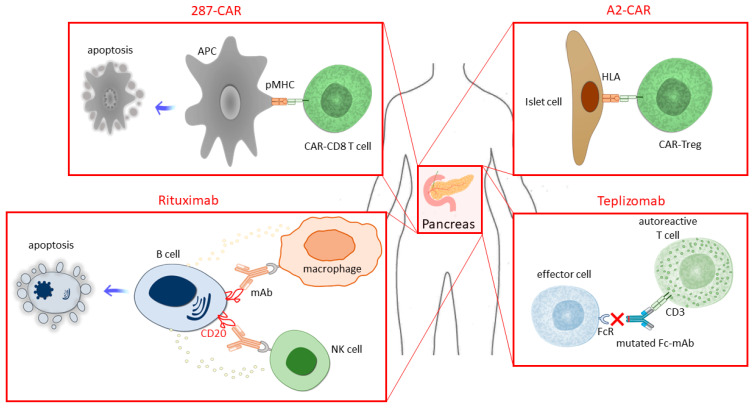
Summary of the most promising strategies for protecting implanted β-cells from immune rejection.

**Table 1 ijms-25-00344-t001:** Summary of the most distinctive CRISPR–Cas nuclease systems [19].

Cas13 nucleases (Type VI)	RNA-guided ribonucleases. Process pre-crRNA into mature crRNA. Collateral activity, allowing gene knockdown, base editing, and nucleic acid detection in mammalian cells.
Cas9 nucleases (Type II)	dsDNA cleavage with RuvC and HNH domains. Requires crRNA and tracrRNA for guide RNA. Mostly generates blunt-end cleavage.
Cas12 nucleases (Type V)	dsDNA cleavage with RuvC domain. Requires crRNA for guide RNA. Generates staggered-end cleavage, often with a T-rich PAM preference.
Cascade–Cas3 (Type I)	Type I CRISPR-associated complex for antiviral defense. Forms R-loop structure with crRNA and target DNA. Cas3 is recruited for specific dsDNA cleavage.
Cas7–11 nucleases (Type III)	Generally, uses several Cas proteins to target RNA. Subtype III-E uses a single-protein effector for recognizing and cleaving target RNA. Examples, such as DiCas7-11, exhibit robust RNA knockdown activity without collateral cleavage.

**Table 2 ijms-25-00344-t002:** CRISPR–Cas9-engineered T1D models.

Target	Outcome	Reference
NOD.Ptpn22^R619W^	Increased IAAs	[13]
NOD.Aicda^−/−^	Delayed T1D development	[6]
PTPN1^−/−^ PTPN6^−/−^	Protected β-cells from cytokine-induced cell death	[28]
NOD.HLA class II variants	Restored pathogenic CD8+ T-cell development and T1D susceptibility	[29]
*Nfkbid* ^−/−^	Accelerated T1D onset	[26]
CNS1^−/−^	Decrease in RORγt+ Tregs	[30]
*Rnls* ^−/−^	Protected β-cells against autoimmune destruction	[31]
NOD.STING^−/−^	Higher incidence of spontaneous diabetes	[32]
NOD.Dusop10^−/−^	Downregulation of type I interferon signature genes in islet cells	[10]
CXCL10^−/−^	Improved SC-islet survival	[33]
Lentivirus thymic transfer	Prevented autoimmunity	[34]
MND promoter	Prevented T1D development	[35]
TCRα	Enhanced the avidity of an antigen-specific Treg cell product	[36]

## Data Availability

Not applicable.

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
