# Peer review of "Applications of Genome-Editing Technologies for Type 1 Diabetes"

_ijms, 2023, doi:10.3390/ijms25010344_

Round 1
Reviewer 1 Report
Comments and Suggestions for Authors
The authors summarize the current knowledge of CRISPR-mediated mouse models of Type 1 Diabetes (TD1). The manuscript provides a comprehensive overview of studies that have utilized CRISPR systems to develop T1D models and treatments, emphasizing the rapidly evolving therapeutic strategies in combating this disease.
In my opinion, the manuscript is very well written and thoroughly describes the generation of TD1 mouse models as well as their use. The manuscript is of interest to the scientific community and will therefore be worth publishing.
I only have some minor suggestions to improve the manuscript.
Minor points:
Line 17: there appears to be a typo in “β?cell”.
Line 115: I’d suggest using “insertion” instead of “addition” as it is much more common in this context.
Line 115-119: I strongly recommend rewriting the description of DNA repair upon CRISPR-mediated double-strand breaks (DSB). There are some misunderstandings in the current manuscript. Nambiar et al. provides a great overview of this topic (DOI: 10.1016/j.molcel.2021.12.026). Briefly, DSBs recruit the endogenous repair machinery and are either repaired by non-homologous end joining pathways, which can result in random insertions and deletions (INDELs) or, if a DNA repair template is available (e.g., another chromosome), by homology-directed repair (HDR) pathways. One of these pathways is homologous recombination, but there are also others. In addition, even though the current citation from Doudna and Charpentier is technically correct, I suggest using a more recent publication like Nambiar et al.
Line 119: to my knowledge, “homologous directed recombination” is not a common term. “Homology directed repair” would be much more appropriate in this context.
Line 350: the quality of Figure 1 seems low in my version of the manuscript. The authors may want to verify the quality in the final version of the manuscript.
Author Response
Dear Reviewer 1,
I am writing in response to the valuable feedback provided on our review titled "Applications of Genome Editing Technology for Type I Diabetes" submitted to IJMS for consideration.
We greatly appreciate the time and effort invested in reviewing our work. Your feedback and expertise has been instrumental in shaping the revised manuscript. In response to the comments raised, we have diligently addressed each point. Below, we outline point-by-point changes made in accordance with your suggestions.
Line 17: there appears to be a typo in “β?cell” This has been amended to ‘β-cell’ (line 12).
Line 115: I’d suggest using “insertion” instead of “addition” as it is much more common in this context. As suggested in below point we rewrote entire paragraph (lines 84–104).
Line 115-119: I strongly recommend rewriting the description of DNA repair upon CRISPR-mediated double-strand breaks (DSB). There are some misunderstandings in the current manuscript. Nambiar et al. provides a great overview of this topic (DOI: 10.1016/j.molcel.2021.12.026). Briefly, DSBs recruit the endogenous repair machinery and are either repaired by non-homologous end joining pathways, which can result in random insertions and deletions (INDELs) or, if a DNA repair template is available (e.g., another chromosome), by homology-directed repair (HDR) pathways. One of these pathways is homologous recombination, but there are also others. In addition, even though the current citation from Doudna and Charpentier is technically correct, I suggest using a more recent publication like Nambiar et al. Thank you very much for this comment as it is a very crucial point in the review. We restructured the review and extended the paragraph on gene editing and CRISPR technologies. We rewrote the description of CRISPR considering the suggested reference. Please find the new description from lines 84–104.
Line 119: to my knowledge, “homologous directed recombination” is not a common term. “Homology directed repair” would be much more appropriate in this context. This has been amended to ‘homology-directed repair’ (line 88).
Line 350: the quality of Figure 1 seems low in my version of the manuscript. The authors may want to verify the quality in the final version of the manuscript. We submitted a high-resolution 300-ppi image. We will inform the editor that this is the one to be used in the manuscript.
We are confident that these revisions have positively addressed the issues highlighted during the initial review process.
Once again, we express our gratitude for your invaluable feedback. Please find the revised manuscript attached for your perusal.
Warm regards,
Meritxell Espino Guarch
Reviewer 2 Report
Comments and Suggestions for Authors
The authors reviewed animal models (mostly murine) for studying type 1 diabetes and current approaches for T1D therapy in their review, but instead of focusing on genome editing technology using CRISPR/Cas systems, a substantial part of the review is taken up by consideration of other therapeutic approaches.
Major suggestions:
1. The authors spend a large portion of the text describing the use of antibodies rather than genome editing technologies, this is not consistent with the current title of the article and the purpose stated by the authors in the abstract.
2. The authors only describe the CRISPR/Cas9 system. It is necessary to analyze the existing literature and find examples of using CRISPR/Cas systems of another type (e.g., V) to edit animals, human/animals beta cells and other cells related to T1D research and therapy. The authors should also look for examples of using base editors, primer editors, or other types of advanced genomic editors.
3. Please provide more detailed statistics on the use of Teplizumab. At what stages of clinical trials is its therapeutic effect evaluated? What were the side effects? I am not convinced that teplizumab is a breakthrough therapy for T1D, it only delays the onset of phase 3 T1D but is not a cure for T1D. If the authors think otherwise, please provide strong arguments in favor of this conclusion.
4. Please, search for literature to discuss in review other animal models (e.g., pigs) used to research and treat T1D.
5. Please move lines 106-117 in the introduction and make them more detailed to more clearly explain what the CRISPR/Cas9 system is, what it can do, how it can be used to alter the genome, and what genomic changes can be made with this system. Also, compare the efficiency with which the CRISPR/Cas9 system makes different changes to the genome.
Minor suggestions:
1. Citations in the text should be formatted according to journal rules, e.g. [1] .... [2,3] .....[4-9].
2. line 68 Please, correct the phrase "the rapid degradation rate of the immunoproteasome", immunoproteasome degrades target proteins and the immunoproreasome itself is a highly stable protease complex.
3. line 106 Awkward phrase "CRISPR-Cas9, initially identified as an immune type system present in bacteria". Strictly speaking, the CRISPR-Cas9 system is not a type of immune system, it is a type of prokaryotic defense systems.
4. Line 107-108 "genome-editing technology applicable for the introduction of novel genetic material" is not the most efficient way to use the CRISPR/Cas9 system - its only function is to find a target and make a DNA cut, it does not itself make genomic changes. In case of classic Cas9 editors, it is the cellular repair systems that make the changes after Cas9 makes the cut.
5. line 116 Please correct the phrase "homologous direct recombination" to "homology directed recombination".
6. Include details of which CRISPR/Cas tools were used to create the corresponding model in Table 1.
7. In the text, discuss in detail how the CRISPR/Cas9 system was used to create mouse models.
8. If information is available, state what challenges the researchers faced and overcame in creating the mouse models.
9. How did the authors validate the mouse models, i.e., how did they prove that the resulting model is relevant to T1D therapy in humans?
10. As you describe each mouse gene model, please discuss the range of molecular functions that the target genes perform. Also discuss the negative aspects of knocking out a particular gene. Finally, suggest how likely it is that, given the possible negative effects, they would be appropriate to be implemented in the treatment of T1D in humans.
11. Lines 247-255. The authors need to describe the engineered TCR gene in more detail to better explain the subsequent changes in CD8+ T cells. Is this a CAR-T technology? State this clearly.
12. Explain in the text how T cells cytotoxic to the HLA-A*0201+ human b-cell lineage will help in T1D therapy?
13. Lines 264-267. Please describe in detail what genomic changes were made to produce pigs compatible with organ transplantation to humans. What do you mean by underdeveloped organs?
14. Describe in more detail the derivation of pancreas from pigs for xenotransplantation to humans including pros and cons.
15. Describe the negative effects of immunosuppression.
16. Lines 332-334 The sentence seems to be incomplete.
17. Figure 1 disrupts the text, please place it after the paragraph.
18. Section 6 should be more structured. Do not mess antibodies with the application of genome editing!
19. In section 6 provide more examples of genome editing approaches being used to protect transplanted beta cells.
Comments on the Quality of English LanguageThere are stylistic errors in the text that mislead readers, need to be corrected. An intermediate level of English corrections is required.
Author Response
Dear Reviewer 2,
I am writing in response to the valuable feedback provided on our review titled "Applications of Genome Editing Technology for Type I Diabetes" submitted to IJMS for consideration.
We greatly appreciate the time and effort invested in reviewing our work. Your insightful comments and constructive criticism have been instrumental in refining our review and improving the overall quality of the manuscript.
In response to the comments raised, we have diligently addressed each point and made significant revisions to the manuscript. Below, we outline point-by-point changes made in accordance with your suggestions.
Major suggestions:
- The authors spend a large portion of the text describing the use of antibodies rather than genome editing technologies, this is not consistent with the current title of the article and the purpose stated by the authors in the abstract. We considered it important to describe the strategies to protect β-cells from autoimmunity as is essential to the success of the transplanted engineered cells. Based on your suggestion and to stay focused on the topic of the review ‘Applications of Genome Editing Technologies for Type 1 Diabetes’ we have restructured the text and expanded on engineered cells to block autoimmunity rather than antibodies. Additionally, with this restructure, we addressed the major suggested points. The final structure of the review is as follows:
-
-
- Diabetes Etiology
- Gene Editing Technologies
- CRISPR–Cas Systems
- T1D in vivo models
- CRISPR–Cas9-Engineered T1D Mouse Models
- β-Cell Replacement Therapies
- Insulin production
- Transplants
- Stem cells and Induced Pluripotent Stem cells therapy
- Strategies for Protecting Transplanted β-Cells
- Immunosuppressants Drugs
- Engineered Cells
-
- The authors only describe the CRISPR/Cas9 system. It is necessary to analyze the existing literature and find examples of using CRISPR/Cas systems of another type (e.g., V) to edit animals, human/animals beta cells and other cells related to T1D research and therapy. The authors should also look for examples of using base editors, primer editors, or other types of advanced genomic editors. Following your advice, we described all types of CRISPR/Cas systems as well as alternative gene editing technologies. Please see lines 84–104 and Table 1.
- Please provide more detailed statistics on the use of Teplizumab. At what stages of clinical trials is its therapeutic effect evaluated? What were the side effects? I am not convinced that teplizumab is a breakthrough therapy for T1D, it only delays the onset of phase 3 T1D but is not a cure for T1D. If the authors think otherwise, please provide strong arguments in favor of this conclusion. We rephrased the sentence and include a broader discussion and data on teplizumab. Please refer to lines 415–419.
- Please, search for literature to discuss in review other animal models (e.g., pigs) used to research and treat T1D. We included all in vivo T1D models, not just mice. We have mentioned non-gene edited models and extended on the rest. Please refer to lines 109–135.
- Please move lines 106-117 in the introduction and make them more detailed to more clearly explain what the CRISPR/Cas9 system is, what it can do, how it can be used to alter the genome, and what genomic changes can be made with this system. Also, compare the efficiency with which the CRISPR/Cas9 system makes different changes to the genome.Thank you for your valuable input. We have changed the structure of the review and the Introduction includes gene editing technologies and a detailed explanation of CRISPR/Cas Systems (as suggested in point 2) as well as amending this point. Please refer to lines 77–104 in the Introduction.
Minor suggestions:
- Citations in the text should be formatted according to journal rules, e.g. [1] .... [2,3] .....[4-9]. The format of the citations has been changed to Arabic numbers in square brackets, as per the journal guidelines.
- line 68 Please, correct the phrase "the rapid degradation rate of the immunoproteasome", immunoproteasome degrades target proteins and the immunoproreasome itself is a highly stable protease complex. The sentence has been amended to "a rapid degradation rate in the immunoproteasome" (line 64).
- line 106 Awkward phrase "CRISPR-Cas9, initially identified as an immune type system present in bacteria". Strictly speaking, the CRISPR-Cas9 system is not a type of immune system, it is a type of prokaryotic defense systems. This sentence has been amended to “identified as a prokaryotic defense system present in bacteria” (line 82).
- Line 107-108 "genome-editing technology applicable for the introduction of novel genetic material" is not the most efficient way to use the CRISPR/Cas9 system - its only function is to find a target and make a DNA cut, it does not itself make genomic changes. In case of classic Cas9 editors, it is the cellular repair systems that make the changes after Cas9 makes the cut. This paragraph had been rewritten. Please see lines 77–104.
- Line 116 Please correct the phrase "homologous direct recombination" to "homology directed recombination". This has been amended to read ‘homology-directed repair’ (line 88).
- Include details of which CRISPR/Cas tools were used to create the corresponding model in Table 1. Table 1 shows mice models created using CRISPR/Cas9 technology. The tile of the paragraph and the table have been amended. Please refer to line 167.
- In the text, discuss in detail how the CRISPR/Cas9 system was used to create mouse models. We have added a section in the Introduction to describe CRISPR/Cas9 systems. Please see lines 77– 104).
- If information is available, state what challenges the researchers faced and overcame in creating the mouse models. Unfortunately, we were unable to identify any findings of this nature.
- How did the authors validate the mouse models, i.e., how did they prove that the resulting model is relevant to T1D therapy in humans? We have summarized the outcomes in Table 1 and added the sentence ‘While essential for preclinical research, these models have limitations and are different to human disease, emphasizing the need for comprehensive studies that integrate findings from animal models with human clinical observations to advance our understanding and treatment of T1D.’ (lines 299–302).
- As you describe each mouse gene model, please discuss the range of molecular functions that the target genes perform. Also discuss the negative aspects of knocking out a particular gene. Finally, suggest how likely it is that, given the possible negative effects, they would be appropriate to be implemented in the treatment of T1D in humans.
For each mouse model, we have described the gene function and the outcome. Unfortunately, we cannot speculate how this could impact if implemented in humans due to a lack of data from translational studies. We have added the sentence “While essential for preclinical research, these models have limitations and are different to human disease, emphasizing the need for comprehensive studies that integrate findings from animal models with human clinical observations to advance our understanding and treatment of T1D.” Please see lines 299–302.
- Lines 247-255. The authors need to describe the engineered TCR gene in more detail to better explain the subsequent changes in CD8+ T cells. Is this a CAR-T technology? State this clearly. This has been amended and extended. Please refer to line 455–477.
- Explain in the text how T cells cytotoxic to the HLA-A*0201+ human b-cell lineage will help in T1D therapy? We have included the sentence ‘Treg cell therapy is a promising approach, but chronic activation by CAR system resulted in the loss of suppressive function by Treg exhaustion and these issues need to be addressed [72].’ Please refer to line 474–477.
- Lines 264-267. Please describe in detail what genomic changes were made to produce pigs compatible with organ transplantation to humans. What do you mean by underdeveloped organs? The section on xenotransplants has been extended. Please refer to lines 348–355.
- Describe in more detail the derivation of pancreas from pigs for xenotransplantation to humans including pros and cons. The section on xenotransplants has been extended. Please refer to lines 348–355.
- Describe the negative effects of immunosuppression. Please refer to lines 410–413.
- Lines 332-334 The sentence seems to be incomplete. This sentence had been amended to ‘Lantidra is administered through a single infusion into the hepatic portal vein and has shown effectiveness in reducing or eliminating the need for insulin in some patients, but adverse reactions in others. This treatment should be considered after assessing the benefits and risks for each individual patient [47].” Please see lines 342–345.
- Figure 1 disrupts the text, please place it after the paragraph. We have relocate the figure to appear after the paragraph. Please see line 422.
- Section 6 should be more structured. Do not mess antibodies with the application of genome editing! The new structure is detailed in point 1.
- In section 6 provide more examples of genome editing approaches being used to protect transplanted beta cells. This section has been extended. Please refer to lines 425–491.
There are stylistic errors in the text that mislead readers, need to be corrected. An intermediate level of English corrections is required.
The manuscript has been edited by professional English language copyeditor.
We believe that these revisions have significantly strengthened the review, addressing the concerns raised and improving its overall quality and readability. We are confident that these revisions have positively addressed the issues highlighted during the initial review process.
Once again, we express our gratitude for your invaluable feedback. Please find the revised manuscript attached for your perusal.
Warm regards,
Meritxell Espino Guarch
Round 2
Reviewer 2 Report
Comments and Suggestions for Authors
The authors have greatly improved their manuscript by responding to all the reviewers' comments. I have no further significant comments and recommend that the revised manuscript be accepted for publication.